# EUS-Guided Radiofrequency Ablation Therapy for Pancreatic Neoplasia

**DOI:** 10.3390/diagnostics14192111

**Published:** 2024-09-24

**Authors:** Mihai Rimbaș, Andra-Cristiana Dumitru, Giulia Tripodi, Alberto Larghi

**Affiliations:** 1Gastroenterology Department, Colentina Clinical Hospital, Carol Davila University of Medicine, 020125 Bucharest, Romania; mihai.rimbas@umfcd.ro (M.R.); andra.cristiana.dumitru@gmail.com (A.-C.D.); 2Digestive Endoscopy Unit, Fondazione Policlinico Universitario A. Gemelli IRCCS, 00168 Rome, Italy; giulia.tripodi91@gmail.com; 3CERTT, Center for Endoscopic Research Therapeutics and Training, Catholic University, 00168 Rome, Italy

**Keywords:** radiofrequency ablation, EUS, EUS-RFA, pancreatic neoplasia

## Abstract

Radiofrequency ablation (RFA) under endoscopic ultrasound (EUS) guidance has been developed and utilized over the last decade to provide the loco-regional treatment of solid and cystic pancreatic neoplastic lesions. The advantage of this approach relies on the close proximity of the EUS transducer to the target pancreatic lesion, which, coupled with the development of specifically designed RFA ablation devices, has made the procedure minimally invasive, with a clear reduction in adverse events as compared to the high morbidity of the surgical approach. EUS-RFA has been applied so far to pancreatic functional and non-functional neuroendocrine neoplasms, pancreatic ductal adenocarcinoma or metastases to the pancreas, and pancreatic neoplastic cysts. Excluding neuroendocrine tumors, for other indications, most of these procedures have been performed in patients who refused surgery or were at high surgical risk. More studies evaluating EUS-RFA in selected patients, not at surgical risk, are gradually becoming available and will pave the road to extend the indications for this therapeutic approach, also in association with other oncological therapies. The present manuscript will critically review the available evidence in the field of the EUS-guided RFA of solid and cystic pancreatic neoplasms.

## 1. Introduction

Therapeutic endoscopic ultrasound (EUS) is now well established and has been a natural evolution of diagnostic EUS, as previously occurred for endoscopic retrograde cholangiopancreatography (ERCP). The proximity of the EUS transducer to the pancreas, coupled with the ease with which a pancreatic lesion can be targeted by a needle, have been the basis for the development of ablative devices for the loco-regional treatment of pancreatic neoplastic lesions. Indeed, surgical resection, which is considered the treatment of choice for pancreatic neoplasia of different degrees of malignancy, carries a significant risk of severe morbidity and mortality, rated in expert centers to be between 16.6% to 19.8%, and 0.6% to 1.4%, respectively [1]. Consequently, the development of alternative minimally invasive therapies has been highly advocated.

Since the first description of EUS-guided radiofrequency ablation (EUS-RFA) in an animal model performed with a prototype device [2], technical advancements have been made and were followed by case reports and case series and more recently by larger studies, mostly for pancreatic neuroendocrine tumors (NETs).

The present paper describes the equipment to perform EUS-RFA and the available evidence on its utilization in neuroendocrine tumors, cystic neoplasms, adenocarcinomas of the pancreas, and pancreatic metastases.

## 2. Description of Radiofrequency Ablation Devices

Multiple devices have been developed to thermally destroy neoplastic tissue under EUS guidance, using radiofrequency current, cryotherapy [3], photodynamic [4], or laser therapies [5]. Among these ablation methods, the most clinical experience has been so far obtained with EUS-RFA, for which two systems are available on the market [6]. The first is the Habib RFA probe (EndoHPB, EMcision UK, London, UK, purchased by Boston Scientific Corp., Marlborough, MA, USA), which is a monopolar 1 Fr (0.33 mm) 220 cm long electrode. The probe can be inserted through a standard 22-gauge fine needle aspiration (FNA) needle (Figure 1), and can be used with commonly available electrosurgical generator units to deliver RF energy. The Habib RFA probe, which is not associated with a cooling system, has been currently retracted from the market for further improvements. The second is the RFA system from Taewoong Medical (Gimpo-si, Gyeonggi-do, South Korea), which comprises the following: (i) the EUSRA needle, which is a 19-gauge, 140 cm long electrode insulated over its entire length, except for the terminal 5 mm to 30 mm that are used to deliver RF energy; (ii) a dedicated radiofrequency generator (VIVA RF generator; Taewoong Medical); and (iii) an inner cooling system that utilizes chilled saline solution circulating through the needle during the RFA procedure (Figure 2).

## 3. Radiofrequency Ablation Procedure

The indications and contraindications of the EUS-RFA procedure are presented in Table 1. Before starting the procedure, antibiotic prophylaxis, as well as intrarectal nonsteroidal anti-inflammatory drugs for the prophylaxis of postprocedural infection and pancreatitis, should be administered [8,9]. Since a close proximity to the main pancreatic duct (MPD) is repeatedly reported as a major risk factor for adverse events (AEs) [10,11], a distance of more than 2 mm between the target lesion and the MPD is recommended to reduce the incidence of postprocedural complications, such as acute pancreatitis or pancreatic duct stenosis. After the application of Doppler ultrasound to exclude intervening vessels, the target area is punctured with the RFA needle. Caution should be taken not to insert the active part of the needle deeper than the lesion, or if this occurs the needle has to be retracted until its tip becomes visible inside the lesion close to its distant margins. Similarly, to avoid the burning of the gastrointestinal wall through which the needle has been passed, which can result in bowel perforation, it is recommended to wait for a few seconds before retracting the device at the end of RF energy delivery.

These precautions being taken, RFA energy is applied using a high-frequency alternating current, to deliver heat at the end of the device and induce coagulation necrosis. For the Habib probe, the manufacturer recommends current application for 120 s (power = 10 W, effect = 4 using a VIO 300D electrosurgical unit, Erbe Elektromedizin GmbH, Tübingen, Germany, in the monopolar soft RF coagulation mode). For the EUSRA device, the producer recommendation is to use a power of 50 Watts, but different powers are generally used depending on the nature of the target lesion and its size, using the VIVA RF generator. The current application has to be stopped when there is an increase in the value of the impedance exceeding 1000 Ω as indicated by the generator [12], which is produced by charred tissue and vaporized water that are electrical insulators [13]. The presence of the device’s inner cooling system avoids sharp increases in heat, allowing the ablation of large volumes of neoplastic tissue without producing tissue charring. Others cease delivering RFA current when the hyperechoic area deriving from ablation sufficiently covers the target lesion or extends beyond the lesion. Thus, real-time control during the EUS-RFA procedure is of paramount importance to avoid potential AEs (see Table 2). Moreover, since artifacts during RFA application may obscure the EUS view of the lesion, it is recommended that RFA be applied first to the farthest and most difficult-to-reach area of the lesion, and to treat the remaining lesion afterward (Figure 3, Figure 4, Figure 5 and Figure 6).

Some authors propose a modification of the generator settings and application times for the EUSRA system currently available on the market, by lowering the RF power (to 30 W) and prolonging the application time until a rise and stabilization in impedance occurs [14]. By using a lower wattage, applied for a longer period of time, a deeper but slower diffusion of the thermal damage is expected, with heat spreading inside (and not outside of) the neoplastic area, making the procedure safer, with less need to relocate the needle to treat other areas, and easier to control in real-time by the endosonographer.

Based on the study by Barret et al. [15], the area treatable with one EUS-RFA shot appears limited. This theoretical limitation can be overcome by giving RF current while retracting the needle from the distal to the proximal portion of the lesion and by repeating RFA ablations within the same treatment session, targeting different areas within the lesion. The need for repeat RFA sessions may largely depend on lesion size. In this regard, contrast-enhanced harmonic EUS (CE-EUS) could be helpful to evaluate for residual viable neoplastic tissue and thus assess the need for further ablation, and to target the residual viable tumor tissue during additional EUS-RFA sessions [16].

## 4. Pancreatic Neuroendocrine Tumors

Pancreatic neuroendocrine tumors (Pan-NETs) represent about 30% of gastroenteropancreatic neuroendocrine tumors (GEP-NETs). In the last decades, the incidence of these tumors has exponentially increased worldwide, mostly due to the widespread utilization of imaging modalities, such as computed tomography (CT) and magnetic resonance imaging (MRI). Indeed, the large majority of Pan-NETs are non-functioning (NF-Pan-NETs) and are detected incidentally when small through imaging studies performed for other indications [17]. The remaining Pan-NETs are characterized by symptoms related to the hypersecretion of a certain hormone, i.e., functioning Pan-NETs (F- Pan-NETs), insulinoma being the most frequently detected one.

Endoscopic ultrasound plays an important role in the localization and diagnosis of Pan-NETs. Localization may be particularly important in the presence of a clinical diagnosis of a hormonal secretion-related syndrome, with normal advanced imaging studies. These tumors can be very small, and EUS has been proven to be superior to both MRI and multidetector CT in the detection of small insulinomas [18]. In these tumors, histological confirmation may not be required, unless the lesion is close or greater than 20 mm in diameter. In this case, the existence of a high proliferation index (Ki-67 > 5%) associated with an increased risk of local and distant metastases should be ruled out before EUS-RFA treatment can be proposed [19]. On the other hand, NF-Pan-NETs require a tissue diagnosis and evaluation of the proliferation index through an EUS-guided fine needle biopsy (FNB), using novel generation FNB needles to better establish further treatment decisions.

The treatment goal is different for F-Pan-NETs and NF-Pan-NETs. For F-Pan-NETs, EUS-RFA should ablate enough tissue to determine the cessation of symptoms by limiting hormonal hypersecretion, because of these tumors’ very low malignant potential. Conversely, in NF-Pan-NETs the entire neoplastic lesion should be ablated. Thus, because of this difference in treatment goals, in patients with functional tumors AEs might be less frequently observed and clinical response rates can be more easily reached than in patients with NF-Pan-NETs.

We ran a search on Pubmed that returned 292 articles on 12 August 2024. Among these, we found 3 meta-analyses, 25 systematic reviews, and 20 original articles (cohort studies, case reports, and case series). By cross-reference, we retrieved six original articles of interest. In Table 3 are presented the original studies referring to the EUS-RFA treatment of insulinomas, and in Table 4 are presented the studies that included patients with different types of pancreatic tumors, including NF-Pan-NETs (excepting case reports).

A recently published systematic review and meta-analysis [9] analyzed 183 patients with F-Pan-NETs (101 lesions, among which were 100 insulinomas) and NF-Pan-NETS (95 lesions). The overall tumor diameter was between 4.5 mm and 30 mm, and most of the patients required one session of EUS-RFA. Only one patient in each group experienced moderate or severe AEs. The study estimated that the rate of AEs was 17.8% (95% CI, 9.1–26.4%) in patients with F-Pan-NETs, and 24.6% (95% CI, 7.4–41.8%) in those with NF-Pan-NETs. Complete ablation (assessed by cross-sectional imaging after the procedure) was obtained in 95.1% (95% CI 91.2–98.9%) of the patients in the first group and in 93.4% (95% CI, 88.4–98.2%) in the NF-Pan-NETs group. This meta-analysis had a few limitations, because not all of the studies included were designed to assess the EUS-RFA efficacy, the inclusion criteria for some of the studies were not recorded, and their follow-up period was relatively short.

Another recently published meta-analysis analyzed 11 studies, excluding those with less than five patients and including a few larger more recent studies, and assessed radiological response after EUS-RFA as the primary outcome [8]. The meta-analysis comprised 292 patients, in whom the pooled complete radiological response after the procedure was 87.1% (95% CI, 80.1–92.8%, I2 = 55.7%), while in 11.4% (95% CI, 6.2–18.1%, I2 = 54.8%) of cases, a partial response was observed. The pooled clinical response for F-Pan-NETs was high, 94.9% (95% CI, 90.7–97.9%, I2 = 0%), most of the functioning Pan-NETs being insulinomas. In patients with tumors less than 15 mm, EUS-RFA had a slightly higher efficacy, as proved by the pooled complete radiological response of 87.9% (81.5–93.0%) vs. 84.3% (61.1–98.0%). The same was observed when an RFA power of less than 50 W was utilized, when the observed pooled complete radiological response was 92.4% (79.8–99.1%) vs. 84.6% (74.6–92.4%) when a power of 50 W was used. A 20% (95% CI, 14–26.7%, I2 = 37.6%) rate of AEs occurred, which were severe in only 0.9% (95% CI, 0.2–2.3%, I2 = 0%) of the cases, rates that seem accurate because most of the studies had a minimum follow-up period of 1 year. No deaths were recorded.

A very interesting study conducted in patients with F-Pan-NETs only, all insulinomas, is a retrospective propensity-matching analysis comparing EUS-RFA and surgical resection. The study found EUS-RFA to be significantly safer than surgery, with lower rates of total (18% versus 61.8%, *p* < 0.001) and severe AEs (0% versus 15.7%, *p* < 0.001) [11], associated with a significantly shorter hospital stay for EUS-RFA patients (3.4 ± 3.0% versus 11.1 ± 9.7%, *p* < 0.001). The clinical efficacy was comparable between the two treatment groups (95.5% versus 100%, *p* = 0.160). Interestingly, among 28 patients in whom the tumor was located less than 2 mm from the pancreatic duct, eight patients developed acute pancreatitis following EUS-RFA. This data suggests that EUS-RFA can become the standard of care for the large majority of patients with F-Pan-NETs, while a multicenter randomized controlled trial is ongoing to directly compare EUS-RFA with surgery as a definitive treatment for these tumors [37].

For small, resectable NF-Pan-NETs without evidence of local or distant metastases at abdominal CT and 68 Ga DOTATATE PET-CT, recent guidelines from the European Neuroendocrine Tumor Society (ENETS) [38] recommended surveillance for lesions with a diameter less than 2 cm, a Ki-67 index no greater than 5% (G1 and low G2 tumors), of an asymptomatic character, with the absence of radiological signs of malignancy, considering patient’s preferences. Conversely, for tumors greater than 2 cm, surgery was the only recommended treatment. No mention of loco-regional treatments was made.

In real life, in two retrospective surgical series, about 30% of patients with NF-Pan-NETs smaller than 2 cm underwent surgical resection, which was associated with grade III AEs in 32% and 18% of the patients, respectively [39,40]. In one of the studies [40], 30- and 90-day mortalities were also non-negligible (2.4% and 3.4%, respectively). A more recent prospective, multicenter study [41] comparing active surveillance (406 patients) vs. surgery (96 patients) in patients with NF-Pan-NETs smaller than 2 cm showed that indication for surgery was directly attributed to patient’s preference in 45% or center’s preference in 39% of the cases, while only in a minority of cases was the indication for surgery determined by the presence of main pancreatic duct dilatation (in 12% of the cases) or distant metastases (in 4%).

Currently, two prospective cohort studies registered on ClinicalTrials.gov are ongoing to assess the safety and efficacy of EUS-RFA in patients with NF-Pan-NETs [42,43]. The size of the lesion to be included differed in the two studies, being between 15 and 25 mm in the RAPNEN study (NCT03834701) and 10 and 20 mm in the RFANET study (NCT04520932). One interesting piece of information is provided by a systematic review of the literature performed by Imperatore et at. [44], who found that according to the ROC curve, a positive response to EUS-RFA was associated with a size ≤ 18 mm of Pan-NETs at EUS. Thus, the current view is that patients with lesions between 20 and 25 mm should be referred to surgery unless they are poor surgical candidates. Lesions less than 14 mm in diameter seem too small to be treated. Indeed, in a two-center study, among 27 treated patients with a mean diameter of 14 ± 4.6 mm, four developed acute pancreatitis, with pancreatic fluid collection formation requiring endoscopic drainage in three of them [31]. All these four patients had lesions no greater than 10 mm. Conversely, in patients with a growing lesion, if EUS-RFA has been chosen as a treatment option, it is important to apply it when the lesion has not become too big to increase the chance of obtaining a complete response.

The major areas of skepticism of oncologists and surgeons about the EUS-RFA treatment of NF-Pan-NETs are the impossibility of verifying the achievement of R0 resection margins and the uncertainty about long-term outcomes. The only available study with more than 3 years of follow-up is limited to 12 patients [27]. Persistence of the complete response was observed in all but one case (91.6%) after a mean follow-up of 45.6 months. The patient with recurrent disease underwent EUS-FNB that disclosed a G1 tumor, for which EUS-RFA re-treatment was proposed but the patient refused. A clear advantage of EUS-RFA is the possibility of repeating the treatment in the case of an incomplete response or recurrent disease at follow-up, while surgery remains an option in the case of failure. This step-up approach can also be utilized in patients with NF-Pan-NETs who need treatment, leaving surgery as a backup for cases of incomplete response or in cases of recurrent disease.

Overall, EUS-RFA appears to be a safe procedure, which can become the standard-of-care for the large majority of patients with F-Pan-NETs. For small NF-Pan-NETs, EUS-RFA seems promising as a minimally invasive treatment alternative to surgery, but data on large cohorts of patients prospectively enrolled are needed to define the exact indications for this procedure and to better report technical data, such as RFA settings and the number of RFA applications.

## 5. Pancreatic Adenocarcinomas

In the United States, pancreatic adenocarcinoma classifies as the third-leading cause of cancer mortality, with a dismal prognosis and a 5-year survival rate of only 11% [45]. Most patients with pancreatic adenocarcinoma have locally advanced or metastatic disease [46], carrying a dim prognosis, for whom systemic therapy is at present the only treatment of choice. For patients with pancreatic ductal adenocarcinoma (PDAC) with a locally advanced disease stage, RFA was initially evaluated during open laparotomy. In a cohort of 100 patients, an increase in survival was found, even though the mortality rate was 3%, mostly related to post-surgical AEs [47]. Subsequently, various studies have evaluated the safety of EUS-RFA in patients with locally advanced or metastatic PDAC.

Table 5 summarizes the available studies on the use of EUS-RFA for PDAC [10,14,48,49,50,51,52,53]. In the first feasibility study on six patients with advanced-stage PDAC [51], EUS-RFA ablation determined the appearance of non-enhancing areas on contrast-enhanced EUS. No major AEs were encountered. Two of the six patients experienced mild abdominal pain that was responsive to analgesic administration.

In another study [14], EUS-RFA was performed on eight patients with unresectable PDAC, all cases considered unsuitable for chemotherapy. CT scan confirmed thermal necrosis of a mean of 30% of the tumor mass (range 5.8–73.5%). Three patients experienced post-procedural abdominal pain rated as mild and responsive to analgesics. No major AEs were observed over a mean follow-up of 4.3 months, such as pancreatitis, biliary or duodenal injury, infection, bleeding, or perforation.

In a third study [53], EUS-RFA was performed in 10 patients with unresectable non-metastatic PDAC, all undergoing neoadjuvant systemic chemotherapy, including additional external radiation therapy in half of them. Upon a CT scan performed at 30 days after EUS-RFA, an intratumoral hypodense area suggestive of necrosis was observed in all cases, with a mean diameter of 30 ± 13 mm. One patient experienced severe abdominal pain during the procedure and underwent concomitant celiac plexus neurolysis, and two additional patients from the cohort experienced self-limiting abdominal pain.

In a small open-label pilot observational study [52], comparing patients with unresectable PDAC receiving standard-of-care chemotherapy with and without EUS-RFA tumor ablation (12 vs. 10 patients, respectively), patients in the intervention group required statistically significantly less narcotic treatment than those in the chemotherapy-alone group, with no difference in the mortality rate at 6 months. Using the EUSRA 19G needle device, 30 procedures overall were performed on the initial 14 patients in the intervention group (median number of procedures of 2.5 per patient; range, 1–4 times), with a mean number of EUS-RFA needle passes per procedure of 5.6 ± 2.9, with a median procedural time of 4.6 min (range, 1.5–16.0 min). One case (7%) of procedure-related mild acute pancreatitis occurred, treated conservatively.

All these experiences proved EUS-RFA to be highly feasible and safe in patients with unresectable PDAC. A few studies also showed that tumor burning from RFA-induced heat and coagulative necrosis released large amounts of tumor antigens, which triggered not only a local but also a distant immune response (the so-called abscopal effect) that could theoretically increase the chance of a response to oncological therapy [6,54]. In an animal cancer model study [55] in which RFA was performed in only one of the two implanted pancreatic cancer sites, tumor growth was found to be significantly inhibited in the RFA-treated tumors. This response was associated with the activation of different antitumor mechanisms, such as an increased expression of C5/C5a, IL23, and CXCL12, along with increased serum levels of chemotactic chemokines CXCL10, CXCL12, and CXCL13, secreted by dendritic cells and myofibroblasts present in the tumor microenvironment, functioning as T and B cell recruiters and promoting the initiation and maintenance of antitumor immune responses. The authors also found that RFA increased neutrophil content and induced tumor microenvironment remodeling in both locally treated and distant non-treated sites, and promoted a systemic antitumor response driven by CD4 and CD8α T cells. This is the first evidence that the RFA-induced systemic effect is capable of limiting tumor progression and inducing stromal and immune modulations, which can be useful in improving the overall therapeutic response.

Thus, the EUS-RFA of PDAC is safe and highly feasible, even though no outcome data are available. Ablation of the tumoral mass seems to stimulate a systemic immune response that acts against the tumor [56] itself, theoretically facilitating the efficacy of concomitant oncological therapy. Further studies utilizing EUS-RFA as part of multimodal oncological treatments, in well-defined patient populations, with full assessment of disease response using the RECIST criteria, are definitely needed to fully explore the value of this ablative therapeutic approach in patients with PDAC [57,58].

## 6. Pancreatic Metastases

Metastases to the pancreas represent a rare etiology of pancreatic tumors. Few case reports published in the literature describe the EUS-guided ablation treatment of metastases to the pancreas. Crinò et al. [14] reported one patient with a pancreatic head metastasis from renal clear-cell carcinoma (RCC) not suitable for surgery or chemotherapy, who was treated with EUS-RFA. By using the EUSRA needle with an active tip of 10 mm, 30 W of power was applied for 55 s, resulting in an ablation of 73.5% of the tumor volume. The patient experienced postprocedural abdominal pain that was managed with analgesics.

In one case series [30], 6 patients with intra-pancreatic metastatic lesions from RCC (10 lesions) or pulmonary carcinomas (1 lesion) were treated by EUS-RFA using the EUSRA needle device with a 10 mm active tip, as part of a cohort of 29 patients. One patient had six pancreatic RCC metastases ranging from 7 to 12 mm, which were all treated in two RFA sessions. Technical success was achieved in all cases. At 6 months follow-up, however, only 2 of the 10 metastatic lesions showed some degree of response. Mild acute pancreatitis occurred in 10.3% of the whole cohort of 29 patients, with no serious AEs observed.

In another study focused on RCC metastases only, EUS-RFA was performed in 12 patients for a total of 21 lesions [59]. The mean size of the RCC metastases was 17 mm (range 3–35), and 33% had an initial size > 20 mm (range 22–35). Because of a systemic metastatic RCC disease, five patients (42%) had received systemic tyrosine kinase inhibitor (TKI) before RFA treatment, while an additional three began a TKI new line after EUS-RFA. Twenty-six EUS-RFA sessions were performed to treat all 21 lesions, with nine patients requiring ≥2 EUS-RFA sessions. No immediate AEs were noted, One patient with an indwelling biliary stent developed hepatic abscesses a few days after EUS-RFA, while in another one a paraduodenal abscess developed two months after EUS-RFA while the patient was on TKI treatment. Overall, after 27.7 months of follow-up, the 6- and 12-month focal control rates were 84% and 73%, respectively. These results made the authors conclude that EUS-RFA is a promising therapeutic option for pancreatic RCC metastases, which needs to be fully evaluated in prospective studies on selected patients. Indeed, selection is of paramount importance, because RCC metastases are usually slowly growing, and probably do not need to be treated when small [60].

In a large national French retrospective study enrolling all EUS-RFA ablative pancreatic procedures performed over a 2-year period, among the 104 procedures performed in 20 centers, 23 involved pancreatic metastases (19 of them from RCC) [10]. The EUSRA needle device with an EUS-RFA power of 50 Watts was used in all cases. A complete response was achieved in seven (30%) of these cases and a partial response in ten (44%), while no response was recorded in six (26%).

Summarizing, the experience gained so far in patients with pancreatic metastases from other tumors confirms the feasibility and safety of EUS-RFA in this clinical setting. At present, a standardized protocol is lacking, and it is still unknown how to include EUS-RFA in a multimodality treatment in this clinical setting.

## 7. Pancreatic Cysts

Pancreatic cystic lesions (PCLs) include a range of entities, each with a different malignant potential. Mucinous cysts, which carry an increased risk for malignancy, account for approximately 61% of PCLs [61] and comprise intraductal papillary mucinous neoplasms (IPMNs) and mucinous cystic neoplasms (MCNs). There could be, however, an overlap in imaging appearance between different entities, which can lead to misdiagnoses. For example, cystic lesions arising from organs adjacent to the pancreas such as duplication cysts or lymphangiomas of the gastrointestinal wall, non-neoplastic pancreatic cysts, renal cysts that can have extrarenal development, or the cystic degeneration of solid tumors can be difficult to discriminate from neoplastic pancreatic cysts [62,63]. In this regard, imaging studies, and especially EUS with the sampling of the cyst content for analysis, play a central role in making the correct differentiation.

For patients with neoplastic PCLs, EUS-RFA has been utilized in a limited number of patients. In a pilot multicenter study [33], six patients with mucinous cysts (4), IPMN (1), and microcystic adenomas (1), who were considered unfit for surgery, underwent EUS-RFA using the Habib catheter. Complete cyst resolution was recorded in two of the cases, while a significant size reduction occurred in the remaining patients after a follow-up period of 6 months. Two of the patients experienced mild abdominal pain persisting for a few days after the procedure. In another study [27] using the EUS-RFA system by Taewoong, 17 patients with pancreatic neoplastic cystic lesions with worrisome features (16 IPMNs and 1 mucinous cystadenoma; 12 with mural nodules of more than 5 mm, 4 with increased wall thickness) underwent EUS-RFA ablation. The mean cyst lesion size was 28 mm, range 9–60 mm. After partial aspiration of the cyst content, the patients received between two to five RFA ablations per treatment session, aiming to determine the filling of the cyst with white bubbles. Two of the patients had a second EUS-RFA session. In one of the first cases, the perforation of an adjacent jejunal loop occurred, requiring surgery. In addition, biliary leakage that was endoscopically managed with drainage and biliary stent placement occurred in another patient seven days after EUS-RFA. No other cases of severe AEs were encountered. A complete response was observed in 11 of the 17 (65%) patients, and one additional patient had a lesion diameter reduction of more than 50% at one-year follow-up. Interestingly, in the twelve cases with mural nodules, RFA treatment determined their complete disappearance in all cases, with no recurrence at one-year follow-up.

The long-term (42.6 months) follow-up of these patients revealed the persistence of the treatment response in 66.6% of cases, with no mural nodules detected. More specifically, in the 15 patients with long-term follow-up, 6 (40%) presented a complete disappearance of the lesion, and 4 patients (26.6%) had a decrease in lesion size of more than 50% (three of them were small recurrences with sizes ranging between 4 and 6 mm). In the remaining five patients, treatment resulted in no change in size or a decrease of less than 50% (three had a stable size, two a slight decrease in size). Interestingly, the development of PDAC occurred in two cases, but was unrelated to the degeneration of the neoplastic cystic lesion and was diagnosed in these patients during follow-up [27].

In a third study, 13 patients with symptomatic pancreatic serous cystic neoplasms or with significant dimensional progression with a median diameter of 50 mm (IQR 34–52.5), who refused surgery or were at high surgical risk, underwent EUS-RFA using the 19 G EUSRA needle and the VIVA RF generator [64]. A contrast enhancement EUS performed one week after the index procedure revealed an incomplete ablation and the need for an additional ablation in six patients. A partial response to RFA treatment was noted in 61.5% of patients, while the rest of the patients had a stable disease after a median follow-up of 9.2 months. Only one patient experienced mild self-limiting abdominal pain with no serologic evidence of pancreatitis. No patients underwent surgery.

In a recent meta-analysis [65], 33 PCLs were identified in four different reports and analyzed as part of a total of 125 pancreatic neoplastic lesions. A complete or partial clinical response was recorded in 24 of the 31 treated lesions (a weighted pooled response rate of 77%). However, an estimation of the overall rate of AEs was not possible, because the data were not presented separately from the other treated lesions, with an overall rate of 3.3%.

Therefore, in neoplastic PCLs, the available evidence is limited and indicates that a significant response should be expected in more than two-thirds of the patients following EUS-RFA ablation, with a low rate of severe AEs. The complete disappearance of the RFA-targeted small mural nodules is promising, but continuous surveillance is required in these patients, because of the risk of developing other high-risk stigmata or pancreatic cancer at other sites.

## 8. Limitations and Future Directions

The main limitation of EUS-RFA is that most of the studies on pancreatic neoplastic lesions are mainly experimental and performed on a limited study population. Despite multiple studies supporting the safety and feasibility of EUS-RFA, the evidence is still very limited, and large randomized studies for determining the appropriate indications and reporting the long-term therapeutic response are still lacking. In this regard, prospective well-designed large controlled studies with an extended follow-up are needed to establish the safety and long-term outcomes of EUS-RFA, including survival.

## 9. Conclusions

EUS-RFA represents a novel minimally invasive therapeutic option for patients with pancreatic solid and cystic neoplasia. In those with pancreatic insulinomas, due to its high safety and effectiveness, EUS-RFA has the potential soon to become the standard of care in the large majority of cases. For the rest of the indications, selection criteria for EUS-RFA treatment still need to be established, while results from studies with better designs performed in a meaningful number of patients are awaited. In this regard, prospective data on a large number of patients, preferably in randomized studies in comparison with current standard-of-care treatments, are needed to better understand the role of EUS-RFA in the multidisciplinary management of patients with pancreatic solid and cystic neoplasia.

## Figures and Tables

**Figure 1 diagnostics-14-02111-f001:**
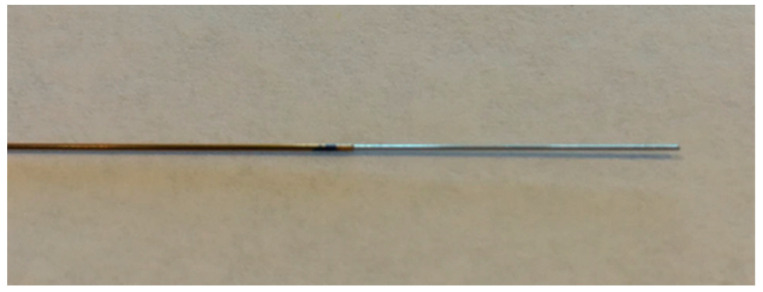
The Habib catheter for EUS-guided radiofrequency ablation (EMcision Ltd., London, UK) (reproduced with permission from [7].

**Figure 2 diagnostics-14-02111-f002:**
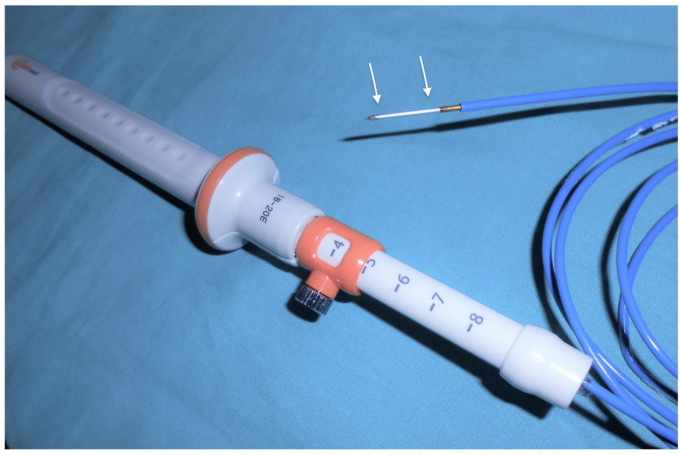
The EUS-guided EUSRA radiofrequency ablation needle from Taewoong Medical with the active tip exposed (arrows) (reproduced with permission from [7].

**Figure 3 diagnostics-14-02111-f003:**
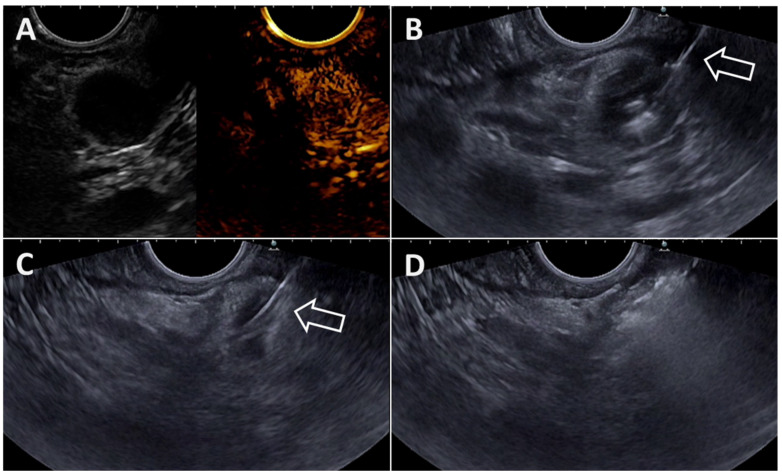
(**A**). Contrast-enhanced endoscopic ultrasound of a 13 mm insulinoma in the uncinate process of the pancreas; (**B**). EUS view of the EUSRA 19G RFA needle (arrow) inserted into the right portion the tumor; (**C**). EUS view of the EUSRA 19G RFA needle (arrow) inside the left portion the tumor. (**D**). Final aspect at the end of the procedure.

**Figure 4 diagnostics-14-02111-f004:**
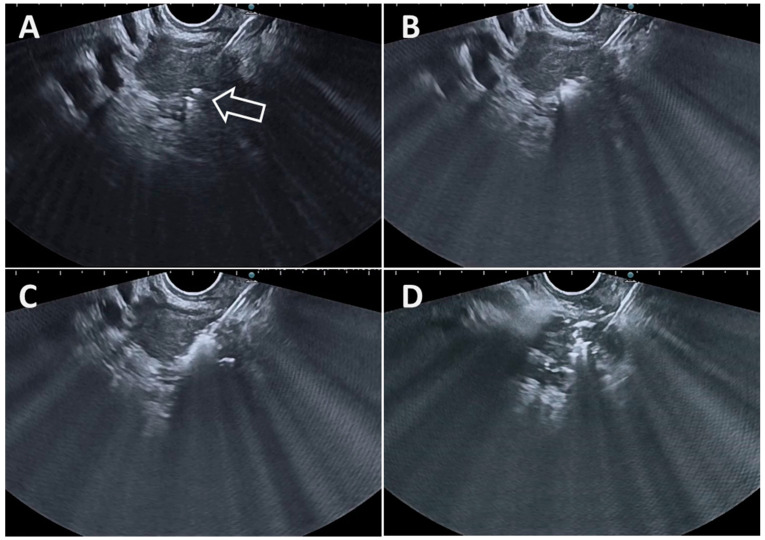
(**A**). Tip of the EUSRA 19G RFA needle inside of a pancreatic head ductal adenocarcinoma (arrow). (**B**). Initiation of the radiofrequency ablation as demonstrated by the appearance of white bubbles. (**C**). Increasing of the ablation site as demonstrated by an increase in the amount of the area with white bubbles. (**D**). Final aspect at the end of the treatment session after targeting three different tumor areas.

**Figure 5 diagnostics-14-02111-f005:**
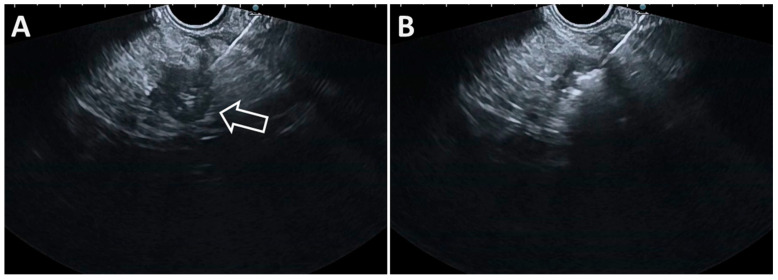
(**A**). The EUSRA 19G radiofrequency device inserted into a pancreatic 15 mm renal cell metastasis (arrow). (**B**). Final aspect after completion of the radiofrequency ablation procedure.

**Figure 6 diagnostics-14-02111-f006:**
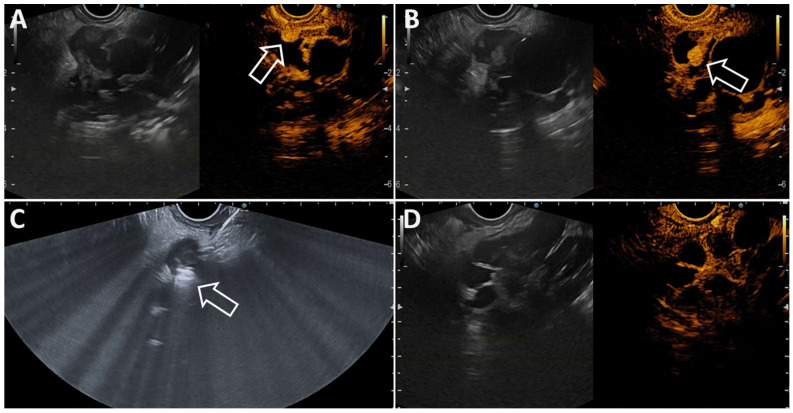
(**A**,**B**). Small mural nodules within a pancreatic neoplastic cystic lesion (arrows) avidly taking contrast as revealed by contrast-enhanced EUS; (**C**). EUS view showing the tip of the EUSRA 19G RFA surrounded by white bubbles (arrow), indicating the beginning of the RFA energy application. (**D**). Follow-up contrast-enhanced endoscopic ultrasound of the same neoplastic cystic lesion after radiofrequency ablation treatment resulting in the disappearance of both mural nodules.

**Table 1 diagnostics-14-02111-t001:** Indications and contraindications of EUS-guided radiofrequency ablation therapy in neoplastic pathology.

**INDICATIONS**	Pancreatic neuroendocrine tumors confined to the pancreas with no distant spread	Tumors with a size between 14 and 25 mm
	Pancreatic adenocarcinomas	Experimental at this stage
	Pancreatic metastases	Experimental at this stage
	Pancreatic neoplastic cysts without suspicion of malignancy, including cysts with worrisome features such as mural nodules, thick septa, or increased wall thickness	Experimental at this stage
**CONTRAINDICATIONS**	Bleeding diathesis that cannot be corrected or anticoagulants that could not be discontinued pre-procedurally	
	Inability to obtain a needle path with no intervening vessels	
	Pancreatic tumor lesion located less than 2 mm from the main pancreatic duct	
	Lack of informed consent for the procedure	

**Table 2 diagnostics-14-02111-t002:** Potential complications of EUS-RFA ablation of pancreatic neoplastic lesions.

Abdominal pain
Bleeding or hematoma
Acute pancreatitis
Gastrointestinal wall perforation
Pancreatic ductal stenosis
Diabetes mellitus
Infection
Miscellaneous (not specified or reported before)

**Table 3 diagnostics-14-02111-t003:** Original studies that apply EUS-RFA in insulinomas.

Author, Year	No.	Age (Years) Men (%)	Tumor Size (mm)	Location	Technical Information		Antibiotic Prophylaxis Rectal AINS	Follow-Up (Months)	Clinical Efficacy	AE (%) Type ٢
					Needle Size (Gauge) Power (W) Impedance (Ohm)	Applications Time (s) No. of Sessions				
Biermann M. et al., 2024 [20]	3	NS (46–90) 33.3%	<20	Head: 33.3% Uncinate: 33.3% Tail: 33.3%	19 20–30 NS	4–10 10–25 NS	NS NS	NS (3–14)	100%	33.3% Mild
Borreli de Andreis et al., 2023 [21]	10	67.1 ± 10 (mean) 30%	11.9 ± 3.3	Uncinate: 30% Body: 30% Tail: 40%	19 20–50 NS	1–14 ≤20 1 (90%) 2 (10%)	NS NS	19.5 (12–59)	100%	20% Mild
Crino S.F. et al., 2023 [11]	89	55.1 ± 16.0 30%	13.4 ± 3.9	Head/uncinate: 38.2% Body: 43.8% Tail: 18.0%	18–19 10–50 NS	NS NS NS	Yes (68.5%) Yes (83.5%)	23 (14–31)	95.5%	7.9% Mild 10.1%) Moderate
Debraine Z.et al., 2023 * [22]	11	65 (NS) 18%	11 ± NS	NS	NS NS NS	NS NS NS	NS NS	26 ± NS	100%	NS NS
Furnica R.M. et al., 2020 * [23]	4	58 (52–82) 25%	12 ± NS	NS	19 50 100	NS NS NS	NS NS	22 ± NS	100%	NS NS
Lakhtakia S. et al., 2016 * [24]	3	NS NS	NS	NS	19 50 NS	NS NS NS	NS NS	NS (11–12)	100%	0% None
Marx M. et al., 2022 * [25]	7	66 ± NS 14.2%	<20	NS	NS NS NS	NS NS 1	NS NS	21 (3–38)	100%	22% Mild 11% Severe
Rossi G. et al., 2022 [26]	3	NS (82–84) 66.6%	NS (9–14)	Head: 33.3% Body: 33.3% Tail: 33.3%	19 30 500	NS (3–4) NS 1 (66.6%) 2 (33.3%)	NS NS	NS (14–27)	100%	33.3% Mild

Numerical data are represented as mean ± standard deviation or median (range). AINS—non-steroidal anti-inflammatory drug; NS—not specified. *—full text not available. ٢—according to the authors.

**Table 4 diagnostics-14-02111-t004:** Original studies that apply EUS-RFA in different types of pancreatic tumors, including Pan-NETs.

Author, Year	Total No. of Patients No. of Patients with NETs	Age (Years)	Men (%)	Tumor Type Tumor Size of NETs (mm)	Technical Information		Antibiotic Prophylaxis Rectal AINS	Follow-Up (Months)	Efficacy in NETs	AE Type ٢٢
					Needle Size (Gauge) Power (W) Impedance (Ohm)	Applications Time (s) No. of Sessions				
Barthet et al., 2021 [27]	29 12	59.9 (45–77)	58	12 NF-Pan-NETs 17 Cystic tumors 13.4 (10–20)	19–22 50 <500	NS 20–45 1–2	Yes Yes	42.9 (36–53)	85.7% ^†^	13.7% NS
Choi J.H. et al., 2018 * [28]	10 8	NS	NS	7 NF-Pan-NETs 2 SPNs 1 Insulinoma NS	NS 50 NS	NS NS NS	NS NS	13 (NS)	70% ^†^	6.2% Mild 6.2% Moderate
De Nucci G. et al., 2020 [29]	10 10	78.6 ± NS	50	5 NF-Pan-NETs 5 Inuslinomas 14.5 (9–20)	19 20 500–600	2–3 10–25 1	Yes NS	NS (12-NS)	100% ^‡^ 100% ^†^	20% Mild
Ferreira M.F. et al., 2022 [30]	29 23	59 (IQR29)	45	10 NF-Pan-NETs 13 Insulinomas 11 Metastatic lesions ˚ 1 ADK 14.4 ± 6.2	19 50 >100	3 (NS) NS NS	Yes Yes	9.5 (IQR 16)	100% ^‡^	40.3% Mild
Marx M. et al., 2022 * [31]	27 27	65 (NS)	52	27 NF-Pan-NETs 14.0 ± 4.6	NS NS NS	NS NS NS	NS NS	15.7 ± 12.2	93%	14.8% AP
Napoleon B. et al., 2023 * [10]	100 64	64.8 ± 17.6	54	64 NENs 23 Metastases 10 IPMN NS	NS NS NS	NS NS NS	NS NS	NS	NS	22% NS
Oleinikov K. et al., 2019 [32]	18 18	60.4 ± 14.4	56	11 NF-Pan-NETs 7 Insulinomas 14.3 ± 7.3	19 10–50 NS	3–10 90–120 NS	Yes NS	8.7 ± 4.6	100% ^‡^ 94.4% ^†^	11.1% Mild
Pai M. et al., 2015 [33]	8 2	65 (27–82)	12.5	6 PCNs 2 NF-Pan-NETs 27.5 ± 17.7	19–22 5–25 NS	4.5 (2–7) 90–120 NS	NS NS	NS (3–6)	NS	25% Mild
Rizzatti G. et al., 2023 * [34]	56 56	NS	NS	24 FPanNETs 32 NF PAN-NETs <25	NS NS NS	NS NS NS	NS NS	NS (6–12)	100% ^‡^ 96.1% ^†^ at 6 M 88.9% ^†^ at 12 M	10.7% Mild 3.5% Moderate
Thosani N. et al., 2018 * [35]	21 3	62 ± 10	61.9	10 Ductal ADKs 2 Insulinomas 1 VIPoma NS	19–22 NS NS	4.6 ± 2.3 NS 1.6 ±0.6	NS NS	5 (NS)	100% ^‡^	2.9% Mild
Younis F. et al., 2022 * [36]	12 7	NS	NS	6 PCNs 1 Insulinoma 6 NF-PanNET 8.9 (6–18)	19 NS NS	4.6 ±2.3 NS NS	NS NS	7 (NS)	100% ^‡^ 66.7% ^†^	25% Mild

Numerical data are represented as mean ± standard deviation or median (range) or median (IQR). AINS—non-steroidal anti-inflammatory drug; NS—not specified; AP—acute pancreatitis; NET—neuroendocrine tumor; NEN—neuroendocrine neoplasm; NF-Pan-NET—non-functioning pancreatic neuroendocrine tumor; F-Pan-NET—functioning pancreatic tumor; VIP—vasoactive intestinal peptide; SPN—solid pseudopapillary neoplasm; ADK—adenocarcinoma; IPMN—intraductal papillary mucinous neoplasm; PCN—pancreatic cystic neoplasm. *—full text not available; ٢٢—according to the authors; ˚—pancreatic and non-pancreatic; ^‡^—clinical efficacy; ^†^—radiological efficacy.

**Table 5 diagnostics-14-02111-t005:** Original studies that apply EUS-RFA in PDAC.

Author, Year	Total No. of Patients	Age (Years)	Men (%)	Tumor Type Tumor Size (mm)	Technical Information		Antibiotic Prophylaxis Rectal AINS Chemotherapy	Follow-Up (Months)	AE (%) Type ٢
					Needle Size (Gauge) Power (W) Impedance (Ohm)	Applications Time (s) No. of Sessions			
Crinò S. et al., 2018 * [14]	9	NS	NS	8 PDACs 1 Renal metastasis NS	18 NS NS	NS NS NS	NS NS NS	6 ± NS	33.3% Mild
Kongkam P. et al., 2023 [52]	14	66.3 ± 10.8	71	14 PDACs 59.7 ± 18.6	19 50 NS	NS NS 2.5 (1–4)	NS NS Yes	NS	7.1% Mild
Napoleon B. et al., 2023 * [10]	100	64.8 ± 17.6	54	64 NENs 23 Metastases 10 IPMNs NS	NS NS NS	NS NS NS	NS NS NS	NS	22% NS
Oh D. et al., 2022 [48]	22	60.5 (IQR, 56.25–68.75)	59	14 Locally advanced tumors 8 Metastatic tumors 38 (32.75–45)	19 50 NS	NS NS 5 (IQR, 3.25–5.75)	Yes NS Yes	21.23 (IQR, 10.73–27.1)	3.7% NS
Robles-Medranda C. et al., 2024 [50]	26	65 (IQR, 56.3–72.8)	53.8	15 LA PDACs 11 M PDACs 39.5 (IQR, 35–43.3)	19 50 NS	NS 5–10 1–3	Yes NS Yes	NS	0% None
Scopelliti et al., 2018 * [53]	10	NS	NS	10 PDACs	18 NS NS	NS NS NS	NS NS NS	NS	0% Major
Song T. J. et al., 2016 * [51]	6	62 (43–73)	NS	NS 3.8 cm (3–9 cm)	18 20–50 NS	NS 10 NS	NS NS NS	NS	33.3% Mild
Thosani N. et al., 2022 [49]	10	62 ± NS	70	7 LA PDACs 3 M PDACs NS	19–22 10–15 200	NS NS 1–4	NS NS Yes	>30	55% Mild

Numerical data are represented as mean ± standard deviation or median (range) or median (IQR). AINS—non-steroidal anti-inflammatory drug; NS—not specified; AP—acute pancreatitis; PDAC—pancreatic adenocarcinoma; NENs—neuroendocrine neoplasms; LA PDAC—locally advanced pancreatic adenocarcinoma; M PDAC—metastatic adenocarcinoma. *—full text not available; ٢—according to the authors.

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
