# Peer review of "EUS-Guided Radiofrequency Ablation Therapy for Pancreatic Neoplasia"

_diagnostics, 2024, doi:10.3390/diagnostics14192111_

Round 1

Reviewer 1 Report

Comments and Suggestions for Authors

The review "EUS-Guided Radiofrequency Ablation Therapy for Pancreatic Neoplasia" presents important information on the management of pancreatic cancer using EUS ablation. Recommendations:

1.        The introduction section should include more information on the general context of the intervention.

2.       Create a flow chart that presents the main indications and contraindications of EUS-Guided Radiofrequency Ablation Therapy in neoplastic pathology.

3.       Unfortunately, the figures are not available for viewing (possibly due to an editing error).

4.       The sequence from lines 121 to 130 should be removed.

5.       In Chapter 4, discuss the importance of EUS in the differential diagnosis of neighboring, often rare, pathologies. I recommend referencing: 10.3390/diagnostics14070675.

6.       In the tables, present only the reference, not the author’s name.

7.       Create a table listing the main complications of the intervention.

8.       Discuss the limitations of the study and future perspectives, ideally in a new chapter.

Author Response

The review "EUS-Guided Radiofrequency Ablation Therapy for Pancreatic Neoplasia" presents important information on the management of pancreatic cancer using EUS ablation. Recommendations:

  1. The introduction section should include more information on the general context of the intervention.

For increasing the readability of the paper, these data were included at the beginning of every of the subchapers “Pancreatic Neuroendocrine Tumors”, “Pancreatic Adenocarcinomas”, “Pancreatic Metastases” and “Pancreatic Cysts”.

  1. Create a flow chart that presents the main indications and contraindications of EUS-Guided Radiofrequency Ablation Therapy in neoplastic pathology.

A Table has been added to the paper, as suggested.

  1. Unfortunately, the figures are not available for viewing (possibly due to an editing error).

The figures have been included in the text file of the revision (not uploaded as separate files), so the reviewers can have them available.

  1. The sequence from lines 121 to 130 should be removed.

The sequence was removed, as suggested.

  1. In Chapter 4, discuss the importance of EUS in the differential diagnosis of neighboring, often rare, pathologies. I recommend referencing: 10.3390/diagnostics14070675.

The discussion has been added, as recommended, including the mentioned reference.

  1. In the tables, present only the reference, not the author’s name.

In all tables the first column’s header has been changed from “Reference” to “Author, year”. We think this is the best way to present the different studies.

  1. Create a table listing the main complications of the intervention.

A table has been added, as suggested.

  1. Discuss the limitations of the study and future perspectives, ideally in a new chapter.

The discussion about the limitations and further perspectives has been added, as suggested.

Reviewer 2 Report

Comments and Suggestions for Authors

1.          This manuscript performed a valuable review about the present aspect of EUS-RFA for the targeted treatment of pancreatic neoplastic lesions, emphasizing the usefulness of EUS-guided approaches as alternatives to traditional surgical methods.

2.          While promising, the available evidence is still limited in terms of long-term outcomes and recurrence rates, particularly for non-neuroendocrine tumors.

3.          The figures were not attached to this manuscript PDF file although the figure legends were provided

Comments on the Quality of English Language

The quality of English language in this manuscript is generally fine.

However, for line 408/409: “Interestingly, in the twelve cases with mural nodules. RFA treatment determined in all cases their complete disappearance, with no recurrence at one-year follow-up.”, this sentence seemed to have some grammatic problem.

Author Response

  1. This manuscript performed a valuable review about the present aspect of EUS-RFA for the targeted treatment of pancreatic neoplastic lesions, emphasizing the usefulness of EUS-guided approaches as alternatives to traditional surgical methods.

Thank you very much for your evaluation.

  1. While promising, the available evidence is still limited in terms of long-term outcomes and recurrence rates, particularly for non-neuroendocrine tumors.

Thank you very much for your evaluation.

  1. The figures were not attached to this manuscript PDF file although the figure legends were provided.

The figures have been included in the text file of the revision (not uploaded as separate files), so the reviewers can have them available.

Comments on the Quality of English Language:

The quality of English language in this manuscript is generally fine.

However, for line 408/409: “Interestingly, in the twelve cases with mural nodules. RFA treatment determined in all cases their complete disappearance, with no recurrence at one-year follow-up.”, this sentence seemed to have some grammatic problem.

The sentence has been modified, as suggested.

Round 2

Reviewer 1 Report

Comments and Suggestions for Authors

The authors have implemented the recommended revisions.